# OpenReview forum: "OODBench: Out-of-Distribution Benchmark for Large Vision-Language Models"
_ICLR.cc/2026/Conference — Submitted to ICLR 2026_

### Official Review · Reviewer_zL11 · 2025-10-18

**Soundness:** 2
**Presentation:** 2
**Contribution:** 3
**Rating:** 6
**Confidence:** 4

**Summary:**

This paper introduces OODBench, a novel and automated benchmark designed to evaluate the performance of large Vision-Language Models (VLMs) on out-of-distribution (OOD) data. The authors propose an effective pipeline to collect 40,000 instance-level OOD pairs by using robust classifiers as OOD detectors. The submission also presents a multi-faceted evaluation metric, the Basic-to-Advanced Progression (BAP), to probe model capabilities in recognition, counting, and reasoning. Through extensive experiments, the paper compellingly demonstrates that even state-of-the-art VLMs suffer significant performance degradation on OODBench, highlighting a fundamental challenge in model generalization.

**Strengths:**

1. The paper tackles the crucial issue of OOD robustness in VLMs, an area vital for real-world applications yet relatively underexplored. It successfully moves beyond simple misclassifications to create a more nuanced benchmark for generative vision-language tasks.

2. The proposed automated pipeline for OOD data collection is a significant strength. The use of multiple detectors in a cross-validation-style approach is a principled method to mitigate the biases of any single detector, ensuring the curated dataset is a more generalized and reliable testbed for OOD performance.

3. The authors conduct a thorough evaluation across a wide and relevant selection of eight VLMs, including recent state-of-the-art models. This broad analysis ensures the findings are generalizable and provides a valuable snapshot of the current landscape of VLM robustness.

**Weaknesses:**

1. While the quantitative results are strong, the paper would be significantly enhanced by a deeper qualitative analysis. Including visual examples of the OOD samples and discussing common failure patterns would provide the community with more intuitive insights into why these models fail.

2. The evaluation focuses on a diverse set of models but misses the opportunity to analyze the effect of model scale on OOD robustness. Including experiments on a model family with varying parameter counts (e.g., small, medium, and large versions of the same architecture) would be highly insightful.

3. The paper's experimental results strongly differentiate OOD from hallucination. However, the qualitative examples presented in the paper (e.g., Figure 4) are paradigmatic cases used to test for hallucination.

4. Discussing the potential boundaries of the OODBench framework—for instance, the types of distributional shifts it primarily captures or its reliance on existing classifiers—would provide a more complete and balanced perspective on the work.

**Questions:**

see above

---

> ### Author Response · Authors · 2025-11-30
>
> 1. In Section 6 of the revised manuscript, we demonstrate how OODBench samples can cause models to fail under various conditions. Additionally, we supplement our analysis with a systematic examination of common failure patterns, including scenarios involving non-main objects and semantic variants. This provides the community with a more intuitive understanding of the fundamental reasons behind model failures under semantic OOD constraints.
>
> 2. We have supplemented Appendix G of the revised manuscript with a systematic analysis of different parameter scales within the same model family. Specifically, we selected the representative Qwen2-VL series. We conducted comparative experiments on its 2B and 7B variants across the whole and subset datasets of the four domains covered by OODBench (COCO, LVIS, nuScenes, Cityscapes). The results reveal no monotonic relationship between model scale and OOD robustness: the 2B model achieved higher accuracy than the 7B model on OOD-H for COCO, nuScenes, and Cityscapes, while the 7B model performed marginally better on LVIS. These mixed results indicate that increasing model parameters does not consistently improve performance under semantic OOD conditions. Furthermore, this trend extends to larger models as well. For instance, models like GPT-4o and Gemini, with significantly more parameters than 7B, do not consistently outperform smaller models on OODBench. Overall, model scaling does not fundamentally resolve semantic OOD challenges. This is because when test samples deviate from the support domain of training data in the joint image-text distribution, their OOD characteristics remain independent of model scale and do not naturally diminish with increased model capacity. We have further emphasized this phenomenon in the revised manuscript and highlighted its significant implications for future research on scaling laws and OOD robustness.
>
> 3. We understand the risk pointed out by the reviewers: In Figure 4, since we employed typical visual comprehension questions such as existence, counting, and comparative reasoning, the format resembles the question templates used in some hallucination literature. This may lead some readers to believe that “this is a hallucination probing mistakenly.” It should be clarified that existence judgments, quantity estimation, and comparative reasoning form the most fundamental and universal sequence of visual tasks. These are widely employed across detection, VQA, referential reasoning, and other domains—not exclusively confined to hallucination research. We selected these questions because they naturally form a progressively challenging task sequence, facilitating systematic evaluation of model robustness across varying complexities—not to simulate any specific hallucination testing scenario. Even if the question formats appear superficially similar, readers can clearly discern the fundamentally different research focus by considering our input design (semantic shifts constructed by OODBench) and experimental objectives. To avoid potential confusion, we have supplemented the main text with more direct comparative explanations to clarify the distinctions between these two task types.
>
> 4. We have supplemented the Limitations section with a more comprehensive discussion, focusing on OODBench's potential reliance on existing classifiers across different domains and the resulting limitations. We also emphasize that the OOD concept proposed in this work is designed for natural images and multimodal semantic learning scenarios. It is not suitable for direct application in highly specialized domains such as medical imaging or remote sensing. In these fields, researchers should redefine appropriate OOD types based on their unique semantic structures, task requirements, and distribution characteristics, rather than simply transferring the definitions from this framework.

---

### Official Review · Reviewer_xm31 · 2025-10-25

**Soundness:** 2
**Presentation:** 2
**Contribution:** 2
**Rating:** 4
**Confidence:** 4

**Summary:**

### Method
- The authors discuss the definition of OOD data for VLMs, and they define two types of OOD data:
    - Objects in images that are neither main objects nor semantically related to the main semantic object;
    - Variants or anomalous forms of target objects.

- Second, they introduce OODBench, a new OOD benchmark for VLMs.
    - They create an automated and efficient OOD data division pipeline with minimal human verification.
    - Along with OODBench, they propose the Basic-to-Advanced Progression Metric
### Results
- Based on OODBench, the authors evaluate SoTA VLMs, and show that current models still struggle with OOD data.

**Strengths:**

- This study explores an interesting question on OOD definition for VLMs.
- They introduce a new benchmark and show insights into current VLMs.

**Weaknesses:**

### Major
- Definition of OOD data for VLMs. The authors define two categories: (1) Objects in images that are neither main objects nor semantically related to the main semantic object;  (2) Variants or anomalous forms of target objects.
    - However, it is difficult to assert that these data types are truly “OOD” for current VLMs. These VLMs are highly likely trained on datasets that already contain such cases to improve their robustness.
    - Second, the definition of OOD is somehow ambiguous. For example, how to tell which objects are the main ones in (1), and what form is anomalous in (2).  In prior OOD settings, class labels are used to clearly separate ID vs. OOD, while the current definition relies on subjective judgments, making it difficult to apply consistently.
    - A possible case is that the data is not OOD but just harder than normal questions.

- Could the authors explain more about how they do the one-shot overlap rate experiments (lines 372-399)?
    - I'm wondering how to use VLMs as model-specific detectors to flag their OOD samples?
    - Are VLM reliable to flag their own OOD samples? Do the authors validate this method on some open-source models like LLaVA?

- Could the authors show some insights or opinions on how to solve the proposed OOD problem?
   - Did the authors try some methods to resolve the OOD problem? Like finetuning the models on the data, changing prompts, or using chain-of-thought?

### Minor
- Some sections of the paper seem slightly redundant. For instance, almost half of the abstract explains the importance of OOD in VLMs, and there is repeated content in the introduction. It may be helpful to refine the structure for better clarity and conciseness.
- The format of citations is not correct. Please try to distinguish between \citep and \citet.
- The format of tables can be improved. For example, the left part of Table 1 is sparse, while the right part is very dense, making it hard to tell the numbers.

**Questions:**

Please see the Weaknesses section.

---

> ### Author Response · Authors · 2025-11-30
>
> **Major 1.1|1.3.** Regarding the questions raised by reviewers in Major Concerns 1.1 and 1.3—i.e., whether OODBench samples might have been trained on VLM and thus fail to constitute true OOD, and whether these samples merely represent "difficult examples" rather than semantically OOD—we provide systematic and verifiable statistical evidence in Appendix B to simultaneously rule out both possibilities. First, at the open-source model level, we rigorously computed the semantic overlap rate between model training sets and COCO-train-OOD divisions constructed under consistent rules, based on auditable training data. This was combined with precise binomial confidence intervals, Bayesian inference, and equivalence testing (TOST) for statistical adjudication. Results demonstrate that this semantic OOD division falls statistically significantly below the 5% overlap threshold, thereby rejecting the possibility that "a substantial portion of the training data covers these samples." This also confirms that the OOD nature of the samples stems from a semantic shift rather than an increase in task difficulty alone. Second, at the closed-source model level, we further validated through significance tests of performance degradation (ID→OOD) and "behavioral signature equivalence" analysis that the performance patterns of closed-source models on OODBench are not only statistically significant but also highly consistent with the degradation features observed in open-source models under scenarios widely recognized as OOD. This result similarly refutes the explanations that "these examples are merely harder versions of normal examples" or that "the closed-source model's training data already sufficiently covers relevant examples." In summary, the analysis of distribution properties on the data side and behavioral response analysis on the model side collectively form a mutually independent and mutually corroborating chain of inference: OODBench samples neither fall within the semantic scope of VLM training data nor represent increased difficulty of ordinary examples, but instead exhibit clear, reproducible semantic OOD characteristics for current MLLMs.
>
> **Major 1.2.** We explicitly provide operational definitions for three types of semantic OOD in this paper. Non-main semantic object refers to: in image-text joint supervision, although the object genuinely exists in the image, it is mentioned infrequently in the training text, resulting in unstable image-text alignment relationships. Semantic variants refer to cases where the semantic category of an object remains consistent (e.g., "skateboard," "horse," "cup"), but its material, stylistic appearance, or construction deviates significantly from the typical visual representations of that category in the training distribution. This causes its appearance to fall in a low-density region of the conditional distribution, leading to a shift in the image-to-semantic mapping (e.g., "a skateboard made of cake"). Anomalous morphology refers to cases where the semantic category remains unchanged, but the object's geometric structure, shape, or spatial arrangement significantly deviates from the visual manifold of natural objects, forming off-manifold samples (e.g., shoes and bags with severe deformation, collapse, or structural abnormalities).
>
> In our workflow design, we first define the semantic scope of OOD based on the above definitions, then perform an initial screening using a general-purpose OOD detector on large-scale data. This initial screening phase identifies potential samples with semantic deviations. Subsequently, manual verification further eliminates false positives and ensures all samples strictly adhere to the OOD definition. Since manual annotation occurs only on samples pre-screened by the model, its limited scale and clear judgment criteria significantly reduce noise from human subjectivity.
>
> Experiments in Appendix B also demonstrate that OODBench exhibits stable, verifiable OOD properties on current MLLMs. This further validates that our OOD classification process is sound and reproducible, without introducing unacceptable noise or compromising semantic OOD consistency due to the limited human verification.

---

> > ### Author Response · Authors · 2025-11-30
> >
> > **Major 2.** In one-shot overlap rate experiments, our workflow is as follows:
> >
> > 1) Input OODBench image-text samples into a VLM (e.g., Qwen2-VL, Gemini, GPT-4o);
> >
> > 2) Use whether the model's natural language output matches the annotation as the operational criterion for "whether the model treats this sample as its own OOD";
> >
> > 3) For each VLM, we counted how many OODBench samples it classified as "its own OOD" using the model's specified method.
> >
> > It is crucial to emphasize that we adopted the model-driven OOD detection approach proposed by Averly et al. and Yang et al.[1, 2], not to evaluate whether "VLMs can reliably label their own OOD samples," but to provide a third-party, model-driven external perspective. This perspective can be used to test whether OODBench still exhibits stable OOD characteristics under a different OOD criterion (i.e., "model error is considered OOD").
> >
> > The underlying intuition is:
> > If OODBench samples indeed exhibit semantic shift at the data level, then even under model-driven criterion, different VLM models are more likely to produce errors on these samples; and this systematic increase in error rates partially reflects OOD characteristics. In other words, we do not define OODBench by model errors, but instead use model-specified methodology as an external control to verify that data-side OOD manifests as consistent cross-model bias on the model side.
> >
> > **Major 3.** We thank the reviewers for their suggestions on addressing OOD challenges. In Section 5.4 of the original paper, we did explore using Chain-of-Thought (CoT) reasoning to mitigate performance degradation caused by OOD data. However, experimental results indicate that CoT fails to improve model performance on OODBench effectively and may even lead to further deterioration in several models. As analyzed in Appendix D, the fundamental cause of this phenomenon lies in the fact that CoT's reasoning chain itself starts from the model's initial interpretation of the input semantics. When the input falls outside the training distribution, the model's initial understanding of OOD samples is often biased. CoT then extends this initial interpretation through a longer reasoning process, amplifying the initial bias, reinforcing false assumptions, and accumulating erroneous reasoning chains. This ultimately leads to further performance degradation rather than improvement. This indicates that in semantic OOD scenarios, relying solely on CoT or prompt engineering is insufficient to address the root problem, as the model's "reasoning starting point" already lies outside the support of the training distribution. In future work, we will explore more systematic, semantically-shifted solutions to enhance the robustness of MLLMs under OOD semantics.
> >
> > [1]. Reza Averly and Wei-Lun Chao. Unified out-of-distribution detection: A model-specific perspective. In ICCV, pp. 1453–1463, 2023.
> >
> > [2]. William Yang, Byron Zhang, and Olga Russakovsky. Imagenet-ood: Deciphering modern out-of-distribution detection algorithms. ICLR, 2024b.

---

> > > ### Author Response · Authors · 2025-11-30
> > >
> > > **Minor 1.** We thank the reviewers for their suggestions regarding conciseness in writing. We understand the reviewers' concern about "information overlap between the abstract and introduction sections," but in this work, fully elaborating the problem background of semantic OOD in VLM is necessary rather than redundant.
> > >
> > > First, the definition of multimodal OOD remains fragmented in the current literature, particularly regarding the concept of "semantic OOD" in the VLM context, where no unified consensus has been established. Therefore, separately articulating the research background and problem motivation in the abstract and introduction helps readers quickly grasp the positioning and innovations of our work, preventing potential misunderstandings of the problem formulation due to contextual gaps.
> > > Second, although both the abstract and introduction address the importance of OOD, they serve distinct functions:
> > >
> > > * The abstract summarizes the entire work and provides a highly condensed motivation;
> > > * The introduction systematically argues, within the literature context, why existing methods cannot address our problem setting.
> > > While the information in these sections is related, their functions are complementary and do not constitute redundant content.
> > >
> > > **Minor 2.** We have thoroughly reviewed the entire text and correctly distinguished and used \\citet and \\citep at all citation points to ensure the citation format complies with the guidelines.
> > >
> > > **Minor 3.** We thank the reviewers for their suggestions regarding the readability of the table. We have carefully re-examined Table 1 and wish to clarify as follows:
> > >
> > > The differing content density between the left and right sections of the table stems from their distinct functional purposes:
> > >
> > > * The left three columns present structural information about MLLM (model name, visual encoder, language model), serving as high-level descriptions, hence naturally containing sparser content;
> > > * The multiple columns on the right (Accuracy, F1, Precision, Recall, MCC, etc.) present numerical results, representing core experimental data, and thus appear more visually concentrated.
> > >
> > > This layout aligns with standard presentation conventions in mainstream VLM and MLLM literature, allowing for a clear distinction between structural and numerical elements. Furthermore, we have thoroughly reviewed the final table formatting multiple times. The current column widths, spacing, and font sizes ensure clear recognition of metrics, maintaining excellent readability without any issues of numerical crowding or legibility.

---

### Official Review · Reviewer_TUbR · 2025-10-29

**Soundness:** 2
**Presentation:** 2
**Contribution:** 3
**Rating:** 4
**Confidence:** 4

**Summary:**

The paper proposes OODBench, an instance-level OOD benchmark for VLMs targeting covariate shift. It uses multiple generalized classifiers (e.g., CLIP, BLIP2) with a “purify” step and cross-validation to label OOD-S (symmetric difference) and OOD-H (intersection), with light human verification. Data spans COCO/LVIS (natural) and nuScenes/Cityscapes (driving).
A Basic-to-Advanced Progression (BAP) metric evaluates existential, counting, and logical comparison questions.
Extensive experiments across open/closed models show large performance drops on OOD-H vs. ID and mixed effects of CoT prompting. The paper also differentiates OODBench from hard samples and hallucination via overlap, variance/correlation analyses, and CoT effects.

**Strengths:**

1. Timely problem with safety relevance: focuses on covariate shifts common in real-world deployment (especially driving).
2. Mostly automated, reproducible pipeline with reasonable cross-detector validation and public release.
3. BAP design offers layered diagnostics (recognition → counting → logic).
4. Broad, careful experimentation and ablations (detector types/number, threshold T); analysis distinguishing OOD vs. hard samples/hallucination is thoughtful.

**Weaknesses:**

1. Conceptual clarity: The paper claims a focus on covariate shift but operationally defines OOD as “misclassified/low-confidence” by generalized detectors. Are misclassified samples assumed equivalent to covariate-shifted samples? Misclassification can stem from ID hard cases, label noise, ambiguity, or prompt wording. Please clarify the formal relationship and provide quantitative checks (e.g., factors like scale/occlusion/illumination) to support covariate-shift attribution.
2. Task-level distinction from hallucination: The yes/no queries (“Does this image contain a truck?”) resemble hallucination benchmarks. Although the appendix argues differences (e.g., CoT helps hallucination but not OOD here), clearer, earlier, and more direct contrasts in the main paper (definitions, data construction, and quantitative comparisons) would avoid confusion.
3. Detector dependence and external validity: While detector replaceability is tested, the approach still inherits biases from detector vocabularies/prompts. Sensitivity to text templates/synonyms and guidance for out-of-domain transfer (e.g., medical/remote sensing) are limited. A brief protocol or pilot in a specialized domain would strengthen generality claims.

**Questions:**

See weaknesses.

---

> ### Author Response · Authors · 2025-11-30
>
> 1. We hereby clarify that we do not directly treat mispredictions from generalized OOD detectors as OOD data. Our process involves two distinct steps: 1. First, we provide an explicit semantic definition of OOD (L82–L84), which is entirely independent of any model behavior. 2. We then employ multiple generalized OOD detectors as candidate discovery tools, solely to screen images from large-scale datasets that may align with the definition. The final OOD classification is always rigorously reviewed by humans based on the definition. Therefore, "model prediction errors" serve only as candidate cues within our framework, not as the decisive criterion. Regarding why CLIP (or other generalized OOD detectors) can serve as candidate discovery tools, we emphasize its theoretical and empirical foundation: Taking CLIP as an example, its large-scale contrastive learning mechanism inherently biases the image-text embedding space toward clustering "main semantic objects/normal forms." Non-main semantic objects, semantic variants, or anomalous forms typically exhibit significant deviations within this space. Thus, when an image is paired with non-main semantic text, CLIP's unstable or misprediction indicates that "the current image-text pair's semantic distribution deviates from its training distribution." We utilize this semantic distance anomaly solely as an auxiliary signal for discovering OOD candidates, never directly defining OOD based on model misclassification. Final OOD samples are only incorporated after satisfying our proposed semantic definition and undergoing human review confirmation. Furthermore, Appendix B provides systematic experiments demonstrating that these finalized datasets pose stable and consistent semantic OOD challenges to existing multimodal large models, independently validating the rationality of our classification strategy.
>
> 2. We fully comprehend the potential risks highlighted by the reviewers: since this task employs existential judgment questions (e.g., "Does this image contain a truck?"), its surface form resembles the question templates used in some hallucination detection literature. This similarity may lead some readers to believe we are conducting "hallucination probing mistakenly." We hereby clarify this point. First, presence-based questions represent the most fundamental and universal task paradigm in visual understanding, widely applied across detection, VQA, referential reasoning, and grounding settings—not confined explicitly to hallucination research. We employ such questions not to replicate hallucination testing scenarios, but because they offer the highest causal interpretability, minimal linguistic noise, and most direct quantifiability when controlling semantic variables (semantic shift). True/False questions can be directly mapped to specific semantic shifts constructed by OODBench (e.g., non-main semantic objects, variants, anomalous forms), enabling clear attribution of performance variations to "whether image semantics are OOD" without interference from language modeling factors inherent in open-ended generation tasks. Second, despite superficial similarities in question formats, task definitions, data construction methods, and research objectives, this study's approach fundamentally differs from hallucination detection. Appendix D illustrates these differences across three dimensions: definition, mechanism, and experimental results. Hallucination evaluation focuses on semantic inconsistencies between image facts and model-generated text, whereas our research aims to assess models' discriminative degradation when confronted with systematic semantic shifts. To avoid potential confusion, we have supplemented the main text with more direct comparative explanations to clarify the distinctions between these two task types.

---

> > ### Author Response · Authors · 2025-11-30
> >
> > 3. We thank the reviewers for their valuable comments regarding cross-domain external validity and detector vocabulary/prompt sensitivity.
> >
> > First, **regarding vocabulary**, we conducted systematic OOD Detector replacement experiments in Appendix E, where four models with distinct vocabularies (CLIP, BLIP2, GroupViT, and LiT) were used. The results demonstrate that, despite significant differences in the lexical composition of these detectors, their performance in the initial screening phase remains largely consistent. This indicates that variations in detector vocabularies do not substantially impact the overall effectiveness of the classification process.
> >
> > **Regarding prompt sensitivity**, we emphasize that within this framework, the OOD detector solely generates candidate samples to enhance data screening efficiency, rather than directly determining the final OOD classification. Thus, prompt influence primarily affects candidate recall (i.e., candidate pool size) and does not alter whether final samples meet our proposed semantic-level OOD definition. The final OOD classification still requires rigorous manual review to ensure that prompt bias does not propagate to the final results. More importantly, this work does not aim to investigate prompt engineering sensitivity issues, but rather to propose a scalable and replaceable OOD data division framework. As long as an effective prompt template exists that can trigger the detector's fundamental category discrimination capability, it can meet the requirements for constructing a candidate pool. Minor variations in prompts will not affect the core objectives and conclusions of OODBench.
> >
> > Here, we clarify that the OOD definition used in this work (non-main semantic objects, semantic variants, and their anomalous forms) does not directly apply to highly specialized domains such as medical imaging and remote sensing imagery. We have explicitly stated this limitation in the "Limitations" section of the original manuscript.
> >
> > **In medical imaging**, lesion regions typically constitute the main semantic objects. Their presentation in publicly available medical datasets is highly concentrated and structured, making it difficult to obtain samples of "non-main semantic objects" that meet our OOD definition. Furthermore, "variants" and "abnormal forms" in medical contexts often correspond to professional-level pathological distinctions. Their identification and annotation require medical expert involvement. While our method relies on human confirmation to some extent after automated pre-screening, this entails high expert review costs in medical settings, making practical implementation challenging.
> >
> > **The remote sensing domain** similarly faces fundamental definitional mismatches. Remote sensing images predominantly feature large-scale aerial perspectives, where visual semantics are primarily determined by landform layout, spatial structure, and texture patterns. These images may lack explicit "main semantic objects." Correspondingly, "variants" and "anomalous features" in remote sensing typically denote local texture variations, material property differences, or imaging condition discrepancies—elements that do not constitute "semantic shifts" in the context of image–text semantic alignment. Therefore, our proposed OOD definition (deviation from the main semantic object) does not correspond to a clearly interpretable OOD type within the remote sensing semantic framework and is challenging to detect stably using existing remote sensing image-text models.
> >
> > For these reasons, directly applying this method to medical or remote sensing domains does not align with their respective semantic structures. In the revised version's Limitations section, we further discuss the cross-domain applicability boundaries of this method and highlight the need to redefine OOD forms based on the semantic constructs of each specialized field—a direction worthy of future in-depth exploration.

---

### Official Review · Reviewer_Yfgv · 2025-10-29

**Soundness:** 2
**Presentation:** 2
**Contribution:** 3
**Rating:** 4
**Confidence:** 3

**Summary:**

This work addresses the lack of a comprehensive benchmark to evaluate the performance of Vision-Language Models (VLMs) when dealing with out-of-distribution (OOD) data.

The paper hypothesizes that existing VLMs are primarily trained on the most common classes of visual data. Based on this assumption, the authors define two types of OOD data from a human perception perspective:

1. Objects in images that are neither main objects nor semantically related to the main semantic object.

2. Variants or anomalous forms of target objects.

To operationalize this, the paper proposes a succinct and efficient OOD data division process and introduces OODBench — a benchmark for evaluating VLMs. OODBench includes:

An automatically constructed OOD dataset.

1. A VLM-oriented OOD data division pipeline.

2. A Basic-to-Advanced Progression (BAP) Metric for evaluation.

3. A complete experimental setup for testing multiple state-of-the-art VLMs.

**Strengths:**

1. The pipeline of this paper is clear and easy to be understand.

2. This work proposes that VLMs has seen normal patterns and various instances and it proposes that VLMs should has a new descriptions which is reasonable.

3. This work proposes a new framework in generating data.

4. The bad performance of various VLMs proposes the hard problems of OODbench. Basic-to-Advanced Progression is reasonable for current proposes of LLM-based VLM.

**Weaknesses:**

1. The core idea of OOD benchmark in to evaluting whether VLM can understand the images in In-or-Out training distribution. This work aims in find the images making the VLMs wrongly predict which is not reasonable.
2. This work proposes a BAP evaluating metric but it doese not show in the the first table.
3. In constructing the OODBench, the pipeline is about reducing human cost but there is still human labor involed which is non-reasonable.
4. For the pipeline in OOD collection, it finds the object which makes the base VLMs (CLIP, BLIP-2) wrongly predict. Thus, the OOD data constructed is hard to tell which contributes the bad performance of VLLMs in benchmark.
5. In generating OOD data with pipeline he proposed, there is a clear bottleneck because of the VLMs used in Fig.3.

**Questions:**

The propose directions of semantic shift and covariate shift is not orginally proposed in your paper while there lack of citation.

Writing problem in line 67-71, which is not clear about "unlike".

In Table1 for evaluating performance, there is division of "Open-source Models" but there is no private model.

Random Chance in Table 1 is non-sense and there is extra line at the top for all the tables.

In the tables, there are some models that performs worse than random guess which worth exploration which makes the performance less reasonable.

---

> ### Author Response · Authors · 2025-11-30
>
> 1. We first clarify a key misunderstanding: **this work never treats "VLM's mispredictions" as the basis for defining OOD**. Our process consists of two steps: 1. First, we provide explicit semantic definitions of OOD (L82–L84), including non-main semantic objects, variants, and their anomalous forms. 2. We then employ multiple generalized OOD detectors as candidate discovery tools, solely to screen images from large-scale datasets that may satisfy the definition. The final OOD classification is always rigorously reviewed by humans based on the definition. Therefore, "model prediction errors" serve only as candidate cues within our framework, not as a decisive criterion. Regarding why CLIP (or other generalized OOD detectors) can serve as candidate discovery tools, we emphasize its theoretical and empirical foundations. Taking CLIP as an example, its large-scale contrastive learning mechanism inherently biases the image-text embedding space toward clustering structures of "main semantic objects/normal variants." Non-main semantic objects, semantic variants, or anomalous variants typically exhibit significant deviations within this space. Thus, when an image is paired with non-main semantic text, **CLIP's unstable or erroneous predictions indicate that "the current image-text pair's semantic distribution deviates from its training distribution." We utilize this semantic distance anomaly solely as an auxiliary signal for identifying OOD candidates, never directly defining OOD based on model misclassifications.** Final OOD samples are only incorporated after satisfying our proposed semantic definition and undergoing human review confirmation. Furthermore, Appendix B provides systematic analyses demonstrating that these finalized datasets pose stable and consistent semantic OOD challenges to existing multimodal large models, independently validating the rationality of our classification strategy.
>
> 2. It should be clarified that BAP is a higher-order measure of reasoning progress designed to characterize a model's consistency of performance during the transition from simple to complex tasks. Therefore, it operates at a different analytical level than the foundational accuracy metrics presented in the first table and is not included in the same table. The relevant BAP experimental results and systematic analysis are detailed in Section 5.5 and Table 3 of the main text.
>
> 3. As described in L118–L122, the objective of this work is to construct an OOD data division tailored for MLLM to address the current lack of semantic-level OOD benchmarks in the community. In practice, our goal is not to propose a fully automated OOD division method, but rather to minimize manual effort while ensuring high division quality. Taking the COCO dataset as an example, the original dataset contains approximately 400,000 image–category pairs. After applying our proposed segmentation workflow, the automated pre-screening phase reduces the number of candidate OOD samples to 5,089, representing approximately 1.27% of the original dataset. Subsequent human review requires confirmation only for this small candidate set. This outcome demonstrates that our method effectively balances efficiency and accuracy by significantly reducing manual annotation costs while maintaining OOD division precision and consistency.
>
> 4. We clarify that we do not directly treat the mispredictions of foundational VLMs (such as CLIP or BLIP-2) as OOD data. In lines 82–84 of the original text, we defined OOD data within the VLM semantic learning scenario as: (1) non-main semantic objects or objects unrelated to the main semantic entity; (2) variants and anomalous forms of objects. Foundational VLM serves only as a preliminary screening tool for potential OOD samples, not as a definitive OOD classification standard. Statistical analyses provided in Appendix B demonstrate that the ultimately classified data exhibit out-of-distribution characteristics relative to VLLM. In summary, the classified data possess clear out-of-distribution properties, and the performance degradation of VLLM on this benchmark reflects its inherent generalization limitations in OOD scenarios.

---

> > ### Author Response · Authors · 2025-11-30
> >
> > 5. The VLM (OOD Detector) used in Figure 3 serves solely as a preliminary screening tool. Its function is to efficiently identify potential OOD candidates from large-scale data, thereby significantly reducing the cost of subsequent manual review. The actual OOD division is still confirmed manually based on our proposed semantic definitions. In the revised version, we further present VLM ablation experiments in Appendix E. The results demonstrate that generic OOD detectors (such as different VLM variants) used for initial screening are replaceable. As long as the detector has reasonable category discrimination capabilities, it can effectively complete the initial screening, indicating that our process does not depend on specific models. Furthermore, systematic experiments in Appendix B validate that our proposed OODBench division process consistently constructs OOD challenges with coherent semantic deviation characteristics for multimodal large models, further supporting the effectiveness and robustness of our method.
> >
> > Q1. The OOD definitions of covariate shift and semantic shift are not core contributions of this paper, but rather serve to illustrate the OOD conceptual framework under the traditional paradigm. As shown in lines 171–178 of the original text, we have provided a complete definition and elaboration in the main body. Additionally, we have supplemented the revised manuscript with corresponding literature citations to clarify that this section serves solely for background explanation and does not constitute a novel contribution to the methodology presented herein.
> >
> > Q2. We thank the reviewers for their attention to the manuscript's wording. We have revised the manuscript accordingly.
> >
> > Q3. 	We thank the reviewers for their attention to detail in the paper. We have added annotations to Table 1 to distinguish between closed-source and open-source models clearly.
> >
> > Q4. We thank the reviewers for their attention to detail in the tables. The “Random Chance” row at the top of Table 1 serves to provide a theoretical baseline, helping readers understand the reference range for each evaluation metric. The purpose of this row is to establish a comparable lower-bound benchmark for subsequent model results, thereby highlighting the magnitude and significance of model performance.
> >
> > Q5. In the revised version, we demonstrate how OODBench samples can cause model failures under various conditions. Additionally, we supplement Section 6 of the main text with a systematic analysis of common failure patterns, including scenarios involving non-main objects and semantic variants. This provides the community with a more intuitive understanding of the fundamental reasons behind model failures under semantic OOD constraints.

---

### Comment · Area_Chair_PsUa · 2025-11-20
**To AI Review**

Dear authors,

I would like to share an important reminder regarding the review process. Recently, we have noticed that some reviewers may be using AI tools to help generate their reviews. This can lead to low-quality or inaccurate feedback, which is unfair to authors who deserve careful and thoughtful evaluations.

To help maintain fairness, I kindly ask for your assistance: If you believe a review you received was partly or fully generated by AI, and you have some evidence (for example: unusual writing style, clear factual mistakes, AI-detector results, repeated generic sentences, etc.), please feel free to contact me directly.

I will review any evidence you provide and, if appropriate, adjust the weight of the reviewer’s evaluation so that it does not negatively affect your submission. Thank you for helping us keep the review process fair and responsible. Your understanding and cooperation are greatly appreciated.

Best regards,

AC

---

### Comment · Area_Chair_PsUa · 2025-11-27

Dear Reviewers and Authors,

As we are approaching the rebuttal deadline, I would like to share a gentle reminder with everyone.

For authors:
If you have not yet submitted your rebuttal, please make sure to do so as soon as possible. Submitting very close to the deadline may reduce the chance for reviewers to read and respond in time, which could affect the discussion phase.

For reviewers:
If a rebuttal has already been submitted for your assigned paper, I encourage you to take a moment to read it and, where appropriate, provide a brief response or update your evaluation. Of course, this is not meant to pressure anyone into changing scores, it is simply to ensure that all reviews remain well-informed before final decisions.

Thank you all for your time and effort in keeping the review process smooth and constructive.

Warm regards,
AC

---

### Meta-Review · Area_Chair_y3FS · 2026-01-06

**Summary:**

This paper introduces OODBench, an automated benchmark designed to evaluate the performance of large Vision-Language Models (VLMs) on out-of-distribution (OOD) data. The authors propose a pipeline to collect 40,000 instance-level OOD pairs by using robust classifiers as OOD detectors. The submission also presents a multi-faceted evaluation metric, the Basic-to-Advanced Progression (BAP), to probe model capabilities in recognition, counting, and reasoning. Through extensive experiments, the paper demonstrates that even state-of-the-art VLMs suffer significant performance degradation on OODBench, highlighting a fundamental challenge in model generalization.

Reviewer concerns included the following:
(1) deeper qualitative analysis of data and failure patterns requested
(2) Model scale (e.g. parameters) relative to OOD robustness is not analyzed.
(3) the paper's experimental results strongly differentiate OOD from hallucination. However, the qualitative examples presented in the paper (e.g., Figure 4) are paradigmatic cases used to test for hallucination.
(4) lack of discussion of weaknesses (e.g. reliance on existing classifiers, distributional shifts covered vs. gaps).
(5) definition of OOD for VLMs as objects that are not main objects or semantically related, or variants or anomalous forms of target objects. The reviewer asserts that VLMs are already trained on such objects, and the definition is further ambiguous. Other reviewers note that OOD should question whether the data is used in VLM training, not whether the VLM makes mistakes in recognizing it.
(6) whether VLMs are reasonable to use as detectors to flag their own OOD samples, and what methods were used to resolve the OOD problem (fine-tuning, prompt engineering, CoT, etc.)
(7) Minor formatting issues with citations, tables, etc.
(8) redundancy of content between sections.
(9) human labor is still required despite the 'automated' pipeline.

**Reviewer Concerns:**

The authors revised their manuscript to address concerns (1), (2), and (4). The authors have a discussion of point (3), which may have been a more continued discussion during the rebuttal period, on the difference between analysis for hallucination and OOD.

The authors contend that their definition of OOD (towards point (5)) is sufficient, though the reviewer may continue to disagree. This may have been an extended discussion in the rebuttal period. The authors address point (6) in an appendix, and address all formatting and redundancy concerns (minor points, (7)-(8)).

Authors address point (9) in their rebuttal, clarifying that their goal is not to propose a fully automated OOD division method, but rather to minimize manual effort while ensuring high division quality, such that subsequent human review requires confirmation only for a small candidate set. Further addressing the OOD vs hallucination question, the authors clarify that they do not directly treat the mispredictions of foundational VLMs as OOD data.

**Reviewer Scores:**

zL11: unchanged (5); authors provide experiments to address the reviewer's main concerns in appendix, but review is already leaning positive.
xm31: unchanged (4); authors provided reasonable arguments towards leaving manuscript unchanged for the reviewer's points, but the argument does not seem sufficient to change the reviewer's mind.
TuBR: unchanged (4); shares concerns with other reviewers on definitions, utility of existing detectors and VLMs.
Yfvg: unchanged (4); reviewer had core concerns about the use of human labor and definition of OOD, which despite the rebuttal may not have been satisfied.

---

### Decision · Program_Chairs · 2026-01-26

Reject